# Utility of the ELISpot Test to Predict the Risk of Developing BK Polyomavirus Nephropathy in Kidney Recipients, a Multicenter Study

**DOI:** 10.3390/vaccines13080796

**Published:** 2025-07-28

**Authors:** Abiu Sempere, Natalia Egri, Angela Gonzalez, Ibai Los-Arcos, María Angeles Marcos, Javier Bernal-Maurandi, Diana Ruiz-Cabrera, Fritz Dieckmann, Francesc Moreso, Néstor Toapanta, Mariona Pascal, Marta Bodro

**Affiliations:** 1Infectious Diseases Department, Hospital Clínic, IDIBAPS, University of Barcelona, 08036 Barcelona, Spain; abiusempere@gmail.com (A.S.); ruizcabreradiana@gmail.com (D.R.-C.); 2Servei d’Immunologia, Hospital Clínic de Barcelona, IDIBAPS, Universitat de Barcelona, 08036 Barcelona, Spain; egri@clinic.cat (N.E.); mpascal@clinic.cat (M.P.); 3Department of Nephrology and Renal Transplantation, IDIBAPS, University of Barcelona, 08036 Barcelona, Spain; agonzalezr2@clinic.cat (A.G.); fjbernal@clinic.cat (J.B.-M.); fdiekman@clinic.cat (F.D.); 4Infectious Diseases Department, Hospital Universitari Vall d’Hebron, 08035 Barcelona, Spain; ibai.losarcos@gmail.com; 5CIBERINFEC, ISCIII-CIBER de Enfermedades Infecciosas, Instituto de Salud Carlos III, 28029 Madrid, Spain; mmarcos@clinic.cat; 6Microbiology Department, Hospital Clínic, IDIBAPS, University of Barcelona, 08036 Barcelona, Spain; 7Department of Nephrology and Renal Transplantation, Hospital Universitari Vall d’Hebron, 08035 Barcelona, Spain; francescjosep.moreso@vallhebron.cat (F.M.); nestor.toapanda@vallhebron.cat (N.T.)

**Keywords:** BK polyomavirus nephropathy, kidney transplantation, ELISpot assay, cell-mediated immunity

## Abstract

**Background**: BK polyomavirus (BKPyV) reactivation is a common complication after kidney transplantation and may result in nephropathy and graft loss. As there is no effective antiviral therapy, management focuses on early detection and reduction of immunosuppression, which increases the risk of rejection. Identifying patients at higher risk remains challenging. Monitoring BKPyV-specific T-cell responses could aid in predicting reactivation. This study evaluated the usefulness of ELISpot to monitor BKPyV-specific cellular immunity before and after kidney transplantation. **Methods**: A prospective multicenter study was conducted between October 2020 and March 2022. ELISpot assays were performed prior to transplantation and two months afterward. **Results**: Seventy-two patients were included, with a median age of 56 years; 61% were men, and 24% had undergone previous transplantation. Nine patients developed presumptive BKPyV-nephropathy. No significant differences were found in donor type, induction therapy, or rejection rates between patients with or without nephropathy (*p* = 0.38). Based on ELISpot results, patients were classified into three groups according to their risk of BKPyV-nephropathy. The high-risk group included those who changed from positive to negative at 2 months post-transplant, representing 40% of presumptive BKPyV-nephropathy cases. Patients who remained negative at 2 months were classified as moderate risk (14.5%), while those with a positive ELISpot at 2 months comprised the low-risk group (0%). In the logistic regression analysis, both the ELISpot risk category [OR 19 (CI 1.7–2.08)] and the use of mTOR inhibitors from the start of transplantation [OR 0.02 (CI 0.01–0.46)] were significantly associated with BKPyV-nephropathy. **Conclusions**: Monitoring BKPyV-specific T cells with ELISpot before and after kidney transplantation may help stratify patients by risk of reactivation. Loss of BKPyV immunity at two months is associated with nephropathy, while mTOR-based immunosuppression appears protective. This strategy could guide personalized immunosuppression and surveillance.

## 1. Introduction

BK polyomavirus (BKPyV) is a double-stranded DNA virus that infects the majority of humans in childhood and remains latent in the urothelium and renal tubular epithelial cells. In healthy individuals, BKPyV is asymptomatic. However, in kidney transplant recipients (KTR), immunosuppressive therapy can trigger viral reactivation, leading to BKPyV-associated nephropathy, which affects up to 10% of infected KTR and is a significant cause of premature graft dysfunction and loss [1]. In the absence of an established treatment, the only recommendation is regular screening of BKPyV-DNAemia, which typically has a low positive predictive value, as well as a reduction in immunosuppressive therapy to restore BKPyV-specific immunity. However, this may result in an increase in donor-specific immune response and subsequent acute rejection [2,3]. Although humoral immunity, as measured by neutralizing antibody titers, has been associated with a reduced risk of BKPyV replication, its predictive value for progression to BKPyV-associated nephropathy remains uncertain [4,5].

In contrast, BKPyV-specific cell-mediated immunity (BKPyV-CMI) has been identified as a key parameter for monitoring the course of viral infection. In recent years, IFN-γ-ELISpot has become a reliable method, easily standardized, with high sensitivity, and has emerged as a preferred method for analysing BKPyV-CMI in patients with active BKPyV replication, as it has demonstrated that the presence of the BKPyV-CMI (positive result) is more likely to result in BKPyV clearance [6,7,8].

Clinically, BKPyV-associated nephropathy is classified as possible, probable, presumptive, or biopsy-proven, depending on viral load thresholds and histological confirmation. These categories help guide diagnostic and therapeutic decisions and were used in this study to define outcome groups.

The pre- and post-transplantation monitoring of BKPyV-specific T cells was proposed by Schachtner et al. as a potential marker for identifying renal receptors at risk of reactivation of BKPyV [9]. While numerous studies have been conducted on KTR using ELISpot, only two studies have included both pre- and post-transplant measurements [9,10], realizing multiple measurements after transplantation. Therefore, we hypothesized that with just two measurements (one before transplantation and another 2 months after the transplant, coinciding with the highest level of immunosuppressive therapy), we might provide adequate information to monitor BKPyV-CMI.

We aimed to evaluate some variables as predictive factors for significant BKPyV infection in KTR and the utility of IFN-γ-ELISpot test for BKPyV before and after kidney transplantation as a valuable marker to predict BKPyV-nephropathy.

## 2. Materials and Methods

### 2.1. Patients

We conducted an observational prospective and multicenter study with recruitment of all patients who underwent kidney transplantation between October 2020 and March 2022 from two university hospitals (Hospital Clinic and Hospital Universitari Vall Hebron, Barcelona) with an active transplantation program. IFN-γ-ELISpot test for BKPyV was performed pre-transplantation and 2 months post-transplantation, and different clinical and microbiological data were collected. These two time points were selected to reflect relevant phases in the evolution of cellular immunity in kidney transplant recipients. The pre-transplant time point represents the baseline immune status, while the second month post-transplantation corresponds to the period of maximum immunosuppressive effect, following induction and early maintenance therapy. This interval has been identified as a window of increased susceptibility to opportunistic infections, including BKPyV, as the full impact of immunosuppression typically manifests after the first month. According to Fishman et al., opportunistic infections are infrequent during the first month post-transplantation, with incidence peaking between months 1 and 6 [11]. Moreover, Schachtner et al. demonstrated that BKPyV-specific T-cell responses assessed within the first 2–3 months post-transplantation were predictive of subsequent BKPyV viremia, supporting the relevance of this time frame for immune monitoring [9].

Patients with either less than two ELISpot analyses or indeterminate results (defined as invalid or uninterpretable assays due to technical issues, such as insufficient spot count in the positive control (<20) or excessive background in the negative control (>10) were excluded. All KTRs were followed up for at least 1 year, and a BKPyV standardized management was performed: plasma BKPyV-DNAemia monthly until month 9, then every 3 months until 2 years or more frequently if allograft dysfunction or an unexplained creatinine rise [2]. Maintenance immunosuppression was based on a combination of mycophenolate mofetil (MMF), sirolimus (an mTOR inhibitor), and tacrolimus (a calcineurin inhibitor or CNI). Corticosteroids were given to all patients and were progressively tapered from a starting dose of 1 mg/kg/day to 5 mg/day at 3 months post-transplantation according to our hospital protocol.

The decision to use basiliximab or anti-thymocyte globulin (ATG) for induction therapy followed the hospital protocol, which is based on donor-recipient compatibility and the nephrologist’s clinical judgment.

Management of BKPyV-DNAemia and BKPyV-associated nephropathy followed a standardized hospital protocol. In cases of BKPyV-DNAemia or presumptive BKPyV nephropathy, initial treatment focused on reducing immunosuppression while maintaining the therapeutic regimen (Level 1). This involved reducing the dose of MMF by at least 50% or tapering CNI to achieve trough levels of <6 ng/mL for tacrolimus. For patients on mTOR inhibitors, trough levels were maintained below 6 ng/mL.

If there was no response to this initial strategy, a modification of the immunosuppressive regimen was implemented (Level 2). This included discontinuation of MMF and consideration of alternative regimens, such as tacrolimus + corticoids (targeting tacrolimus levels of 5–6 ng/mL), tacrolimus + sirolimus (3–5 ng/mL and 5–6 ng/mL respectively), or complete conversion to sirolimus + corticoids (sirolimus > 5 ng/mL). In refractory cases (Level 3), advanced therapies, such as donor-specific T-cell infusions or intravenous immunoglobulin, were considered following a multidisciplinary evaluation.

### 2.2. Definitions

Definitions have been revised to be consistent with descriptions in the most recent BKPyV guidelines 2024 [2]. Possible BKPyV nephropathy was defined as high urinary BKPyV load (DNAuria > 10 million copies/mL) but undetectable plasma BKPyV DNAemia. On the other hand, probable BKPyV nephropathy was defined as plasma BKPyV DNAemia >1000 c/mL sustained for > 2 weeks, while presumptive BKPyV nephropathy was defined as plasma BKPyV DNAemia >10,000 c/mL. Finally, biopsy-proven BKPyV nephropathy is defined as evidence of compatible cytopathic effects plus immunohistochemistry and a specific diagnostic test that identifies BKPyV performed upon identification of renal dysfunction or patients with >10,000 c/mL. BKPyV viral load was measured by real-time PCR (ELITech Group, Nanogen, Italy).

Viral infections (Cytomegalovirus or Epstein-Barr virus) were recorded if they occurred at any time during follow-up, regardless of their temporal relationship with BKPyV. CMV viral load monitoring was performed by real-time PCR (ELITech Group, Nanogen, Italy).

Assessment of donor-specific HLA antibodies (DSA) was conducted employing the Luminex-based bead assay technique. The samples were analyzed with the Lifecodes LifeScreen Deluxe kit (Lifecodes, Immucor, Stamford, CT, USA). When the screening yielded positive results, we determined the specific HLA antibody types using the same assay. An MFI greater than 3000 was considered a positive result.

Lymphopenia was defined as a total lymphocyte count <1.0 × 10^9^/L. CD4+ T-cell counts were also recorded at 2 months post-transplantation in all patients.

### 2.3. T Cell Responses Measurement by IFN-γ ELISpot

To determine the presence of T cell responses against BKPyV, PBMC (freshly processed and isolated by Ficoll–Paque density gradient centrifugation) at a concentration of 2 × 10^5^ in a volume of 200 μL, were stimulated in X-VIVO™ 15 medium (Lonza. Basel, Switzerland) with PepTivator^®^ BKPyV LT (1 µg/mL, Miltenyi Biotec, Bergisch Gladbach, Germany) and PepTivator^®^ BKPyV VP1 (1 µg/mL, Miltenyi Biote, Bergisch Gladbach, Germany) peptide pools. These two antigens were selected based on their strong immunogenicity and ability to induce robust virus-specific polyfunctional T-cell responses, which are essential for controlling BKPyV replication [12,13]. Moreover, the use of only LT and VP1 allows for a more efficient and cost-effective assay, reducing the need for additional antigenic pools and the number of PBMCs required—an important factor in immunosuppressed kidney transplant recipients [14].

Negative control wells lacked peptides, while positive control wells included anti-CD3-2 mAb. Cells were incubated for 16 to 20 h at 37 °C 5% CO_2_ in precoated anti-IFN-γ MSIP white plates (mAb 1-D1K, Mabtech, Stockholm, Sweden). After incubation, plates were washed five times with PBS (Sigma-Aldrich) and incubated for 2 h at room temperature with horseradish peroxidase (HRP)-conjugated anti-IFN-γ detection antibody (1 mg/mL; clone mAb-7B6-1; Mabtech). After five further washes with PBS, BCIP/NBT-plus substrate was added, and spots were counted using an automated ELISpot Reader System (Autoimmun Diagnostika GmbH, Straberg, Germany). In order to quantify positive peptide-specific responses, spots of the unstimulated wells were subtracted from the peptide-stimulated wells, and the results were expressed as Spot Forming Units (SFU)/2 × 10^5^ PBMC. We determined BKPyV-specific spot by spot increment defined as stimulated spot numbers ≥10 SFU/2 × 10^5^ PBMC in at least one of the two antigens (VP1 and/or LT). Consequently, we consider a patient positive when they show a specific T cell response (≥10 SFU/2 × 10^5^ PBMC) against one or both antigens.

Each ELISpot assay was performed in duplicate wells for each condition (peptide-stimulated, negative control, and positive control) to ensure reproducibility and consistency of results.

### 2.4. Statistical Analysis

Categorical variables were described as counts and percentages, whereas continuous variables were expressed as either median or interquartile ranges (IQRs). Categorical variables were compared using either a chi-squared test or Fisher’s exact test when appropriate, and quantitative variables, with the Mann–Whitney U test or the t-student test depending on their distribution. Kaplan–Meier curves were constructed for survival analyses. The threshold for statistical significance was defined as a two-tailed *p* < 0.05. All analyses were performed using SPSS software (version 25.0; SPSS, Inc., Chicago, IL, USA) and R (version R-2.13.0R: A language and environment for statistical computing. R Foundation for Statistical Computing, Vienna, Austria).

### 2.5. Ethical Considerations

Procedures were performed in accordance with the ethical standards laid down in the Declaration of Helsinki as revised in 2024. The protocol for this study was approved by the Clinical Research Ethics Committee of the Hospital Clinic of Barcelona (HCB/2019/0925). Written informed consent was obtained from all included patients.

## 3. Results

### 3.1. Demographic and Baseline Clinical Characteristics

During the study period, a total of 392 patients underwent kidney transplantation at the two centers. Due to organizational constraints, including the requirement to process fresh blood samples for ELISpot within a short time frame, only a subset of patients could be actively recruited. Sample processing was limited to working days (Monday to Thursday) and required the presence of a trained technician, whose availability was reduced during certain periods due to scheduled absences. These conditions, which affected both centers equally, restricted the number of patients that could be included at the predefined timepoints. As a result, 111 patients were initially enrolled. Of these, 39 were further excluded: 37 due to the absence of a post-transplant ELISpot (mainly because the sample could not be collected within the required timeframe), and 2 due to loss to follow-up. A total of 72 patients were finally included in the study. A comparative univariate analysis between the screened cohort and the final study population is presented in the Appendix A, finding no significant differences between the two populations. 

Figure 1 describes a diagram of patient enrollment and selection criteria. The mean age was 56 years (SD 13), and 61% were male. The main etiology of end-stage renal disease was diabetes mellitus (29%). Re-transplantation was performed in 17 patients (24%), and there was a total of seven renopancreas transplants (10%). The cause of graft loss was chronic cellular rejection in 10 cases (2 of them with associated thrombotic microangiopathy), chronic antibody-mediated rejection in 2 cases, and a combination of chronic cellular and antibody-mediated rejection in 5 cases. None of these cases was due to BKPyV-associated nephropathy. Furthermore, kidney transplantation mainly came from deceased donors (81%). Induction therapy with basiliximab was administered in 35/72, 50% of cases, while anti-thymocyte globulin (ATG) was used in 36/72, 49% of cases, and rituximab in 3/72, 4% of cases. Some patients received more than one induction agent, most commonly a combination of ATG and basiliximab, depending on immunological risk. Regarding baseline immunosuppressive therapy, 48 patients (66%) were on a calcineurin inhibitor-based regimen while 24 patients (33%) were on an mTOR-based regimen. Main baseline characteristics are summarized in Table 1.

### 3.2. Clinical Outcomes

During the study period, nine patients presented presumptive BKPyV-nephropathy according to BKPyV-DNAemia, with a median time to the first detection of 63 days post-transplantation (SD 148 days). Among these, seven patients underwent allograft biopsy, and three of them had had biopsy-proven BKPyV-nephropathy. Additionally, three patients developed acute allograft rejection, all occurring approximately 3 months post-transplantation. In two of these cases, BKPyV infection preceded the rejection by 1–2 months, while in the third case, BKPyV infection and rejection occurred simultaneously. The severity of the rejections was classified according to the Banff criteria: one was classified as a grade II acute cellular rejection, and the rest as a grade I. Among the 63 patients without evidence of nephropathy during follow-up, 4 developed low-level BKPyV-DNAemia (<10,000 copies/mL) that resolved spontaneously without intervention within 2 weeks. All four patients belonged to the ELISpot-positive group. No significant differences were found between the groups in terms of type of donor (living or deceased), induction therapy, or acute allograft rejection rates (*p* = 0.38). Similarly, no significant differences were observed in total lymphocyte counts at 2 months post-transplantation. CD4 cell count also showed no significant differences between the groups.

During the follow-up, 34 patients experienced detectable CMV-DNAemia (irrespective of BKPyV status). However, only four of them had CMV and BKPyV replication detected simultaneously. No significant differences were observed between the groups regarding the presence of CMV infections or their temporal relationship to BKPyV. There were no Epstein-Barr virus infections nor opportunistic infections (tuberculosis and invasive fungal infection). No patients died during follow-up.

### 3.3. ELISpot-Based Patient Stratification and Risk Analysis

Based in ELISpot results, patients can be classified in three groups: (1) those with a positive ELISpot who became negative (6.9%); (2) those who remained negative at 2 months (66.7%) and (3) those with a positive ELISpot at 2 months (26.4%), regardless of pre-transplant results. Within the first group, two patients (40%) presented with presumptive BKPyV-nephropathy. Similarly, in the second group, there were seven patients with presumptive BKPyV-nephropathy (14%), three of whom developed biopsy-proven BKPyV-nephropathy (44%). In contrast, no patients in the third group manifested BKPyV-nephropathy or DNAemia.

To better illustrate the temporal dynamics of BKPyV-nephropathy occurrence, a Kaplan–Meier survival analysis was performed (Figure 2). Patients in Group 1 (ELISpot +/−2) showed the lowest probability of remaining free from nephropathy over time, while no cases were observed in Group 3 (ELISpot +/+ or −/+). The difference between groups was statistically significant (log-rank test, *p* = 0.011).

In addition, a box plot analysis of quantitative ELISpot values for the VP1 and LT antigens was performed (Figure 3).

We performed a subanalysis of the 24 patients receiving immunosuppressive therapy with mTOR inhibitors. The median dose was 2 mg/day (range: 1–3), and the median level was 2.95 ng/mL (range: 1.6–6.5). Among them, four patients (16.7%) were classified as low-risk (Group 3), 17 (70.8%) as medium-risk (Group 2), and 3 (12.5%) as high-risk (Group 1). Only one patient receiving mTOR inhibitors developed BKPyV-nephropathy, and this patient belonged to Group 2. Figure 4 describes the incidence of BKPyV-nephropathy according to risk groups and mTOR-based immunosuppressive regimen.

Furthermore, a multivariate analysis was performed on the main variables that could be influencing BKPyV-nephropathy (Table 2). The analysis identified the change in ELISpot from positive to negative as a risk factor (*p* = 0.016). In contrast, the use of mTOR treatment from the start of the transplant was found to be a protective factor (*p* = 0.014).

Taking into account only the result of the ELISpot at 2 months, no patient with a reactive ELISpot at 2 months post-transplantation presented nephropathy compared with the non-reactive group (0% vs. 17%, *p* = 0.052). Nevertheless, this association was not statistically significant in the multivariate analysis.

We performed a post-hoc power analysis. For the comparison between reactive ELISpot and non-reactive ELISpot patients at 2 months (0% vs. 17% incidence; n = 19 vs. 53), the achieved power was 88.8% (>80% con, α = 0.05). This suggests that the study was adequately powered to detect clinically meaningful differences, and the borderline *p*-value (*p* = 0.052) likely reflects a real effect limited by the size of the study.

Based on these risk patterns, we propose a clinical algorithm for patient management (Figure 5).

## 4. Discussion

In this prospective multicenter study, we found that 12.5% of patients presented presumptive BKPyV-nephropathy, and 44% of them had biopsy-proven BKPyV-nephropathy. Importantly, we defined a risk index according to the changes in the ELISpot test during the post-transplantation early period that could help clinicians to avoid and manage BKPyV infection.

First of all, we found a high incidence of BKPyV-nephropathy. Based on the most recent registry data, 1–10% of KTRs develop BKPyV-nephropathy [15,16,17]. In line with these results, we have recently seen an increase in the incidence of BKPyV-nephropathy in our centre, which has been related to the increase in high immunological risk transplants, with the use of very intense immunosuppressive regimens at induction, also concordant with the present study, where all recipients received immunosuppressive induction therapy, some with more than one drug.

Secondly, we found that the comparison of BKPyV-CMI status before and after transplantation may provide insights into the behaviour of BKPyV and the probability of replication. While there are nine studies [7,9,10,18,19,20,21,22,23] on the use of ELISpot as a predictive marker in BKPyV, only two of them analyse pre- and post-transplant outcomes [9,10]. Schachtner et al. highlighted the importance of BKPyV-specific T cells in the pre- and post-transplant periods at 30 days in a sample of 24 KTR with BKPyV replication (16 of whom were studied using ELISpot) [9]. Moreover, the study stratified the risk for later infection into four categories according to ELISpot results at baseline and +30 days post-transplantation. Similar to our study, they found that individuals who initially presented with BKPyV-specific T cells but subsequently experienced a decrease or loss were at the highest risk and could potentially benefit from monitoring immunity and reducing immunosuppression in order to control the infection, whereas patients that presented reactivity to ELISpot test at +30 days post transplantation should undergo regular monitorization regardless baseline ELISpot. Finally, Schachtner et al. suggested that further studies are needed to predict the risk of BKPyV replication in patients with undetectable immunity at baseline and at +30 days post-transplantation [9]. Our results support this assumption, showing that this group of patients has an intermediate risk of developing BKPyV nephropathy.

We classified patients into three groups (+/−, −/−, and −/+ or +/+) instead of two because patients in group 1 exhibit a higher risk compared to those in group 2, as reflected by the Kaplan–Meier analysis and Figure 4 (incidence of 40% vs. 14.5%). We believe this classification makes more sense than combining groups 1 and 2. Furthermore, although statistical differences remain if the two groups are combined, we consider it conceptually important that patients who lose cellular immunity should be considered at higher risk than those who do not, based on the baseline analysis. This suggests a more severe degree of cellular immunosuppression.

In contrast, Mutlu et al. demonstrated that pre-transplant BKPyV-specific CD4+ T-cell responses did not show a significant association with subsequent BKPyV reactivation [10]. However, post-transplant monitoring revealed a significant negative correlation between BKPyV-DNAemia and CD4+ T-cell responses, which might provide better guidance for managing the virus. Nevertheless, the study had a small sample size (31 patients with eight viral replications), which limits the generalizability of the findings.

Interestingly, our findings also demonstrated that recipients on mTOR-based regimens had less BKPyV nephropathy. Hirsch et al. showed that sirolimus impaired BKPyV replication in renal tubular cells by inhibiting p70-S6 kinase, suggesting a role for mTOR in early viral replication [24]. The TRANSFORM study found fewer viral infections in the mTOR group compared to mycophenolate (17.2% vs. 29.2%, *p* < 0.001) [25]. A meta-analysis also reported reduced CMV and BKPyV infections in patients receiving mTOR inhibitors [26]. Additionally, mTOR inhibitors enhance CD8+ T cell responses and memory formation [27]. However, the BKEVER trial, a randomized study, did not show greater BKPyV clearance with everolimus versus MMF reduction [28]. The MMF group even had faster viral load decline. Differences in study design, timing of mTOR initiation, or patient selection may explain this discrepancy. Still, mTOR-based regimens might remain a strategic option to reduce opportunistic infections.

Another important finding of our study was that neither acute allograft rejection nor lymphopenia could be considered as risk factors for significant BKPyV infection. While we found that previous acute rejection is more frequent in patients with significant BKPyV infection, we thought that this could be confounded by increased immunosuppression due to anti-rejection treatment. Furthermore, although the use of lymphocyte-depleting agents is classically associated with BKPyV-nephropathy, we did not observe this association, even looking at the number of total lymphocytes or CD4+ lymphocytes.

Finally, no association has been found between Cytomegalovirus and BKPyV. We initially hypothesized that due to the indirect effect of Cytomegalovirus as an immunomodulator, it could facilitate the replication of the BKPyV [29]. While the potential interaction between these viruses is controversial, we could screen for other viruses, such as Torque teno virus, that have been proposed as surrogate markers of profound immunosuppression [30].

Given that cellular immunity monitoring is becoming integrated into clinical practice, as reflected in the most recent 2025 CMV guidelines [31] and that such an approach is not yet incorporated into the 2024 international BKPyV guidelines, we believe that immune-based stratification could help close this gap. Based on our findings, we propose a simplified clinical algorithm to guide both PCR surveillance and immunosuppression adjustment, tailored to each patient’s BKPyV-specific T-cell response at two months post-transplant. In this revised model, we suggest adjusting the frequency of BKPyV-DNAemia screening according to immunological risk: (A) In moderate- and high-risk patients (ELISpot −/− or +/−), closer monitoring is warranted, with PCR every 2–4 weeks, particularly during periods of peak immunosuppression. This risk-adapted strategy may improve resource allocation while maintaining clinical safety and anticipatory management. (B) In low-risk patients (ELISpot +/+ or −/+), who retain BKPyV-specific cellular immunity, PCR monitoring could be safely spaced to every 1–2 months during the first 9 months, instead of monthly as currently recommended.

Nonetheless, there are some limitations that are worth mentioning. First, we included a small number of patients, although a higher percentage of them presented with BKPyV nephropathy. Moreover, there was a lack of BKPyV serostatus pre- and post-transplantation, which could provide us with more information, particularly in primary infections. However, recent data suggest that the usefulness of BKPyV serology in predicting post-transplant infection may be limited. In a recent study by Hillenbrand et al., higher antibody levels were associated with a lower risk of BKPyV-DNAemia, but serology alone did not clearly identify which patients would go on to develop significant viral replication or nephropathy. These findings support the idea that cellular immune monitoring, such as ELISpot, may be more informative for risk stratification in this setting [32]. ELISpot was performed by using LT and VP1 antigens (main immunodominant antigens capable of generating specific CD4+ and CD8+ responses). Although maximum sensitivity to detect T cell responses to BKPyV can be achieved through assessment of the full range of antigens, Chakera et al. showed that in patients who had cleared BKPyV clinically, there was a correlation between the responses to VP1 and large T antigens, but no correlation with responses to any other antigens (VP2, VP3, St) [20].

However, it is important to emphasize the need for methods that support clinical decision-making while considering feasibility and cost-effectiveness. Including all viral antigens would likely be impractical, especially given the limited availability of PBMCs in these patients. Despite its promising predictive value, the ELISpot assay’s cost and complexity may limit its implementation in resource-limited settings. Future cost-effectiveness studies are needed to support its broader clinical adoption.

## 5. Conclusions

Testing BKPyV-specific T cells by ELISpot before and 2 months after transplantation should be a reliable marker for stratifying its risk, and the use of mTOR inhibitors could protect against BKPyV-nephropathy. This approach should be a useful strategy to tailor monitoring intensity and adjust immunosuppressive therapy proactively, particularly in those patients who manifest a loss of BKPyV-CMI at 2 months post-transplantation.

However, further studies are needed to validate these findings and to determine the best way to integrate immune monitoring into routine clinical practice.

## Figures and Tables

**Figure 1 vaccines-13-00796-f001:**
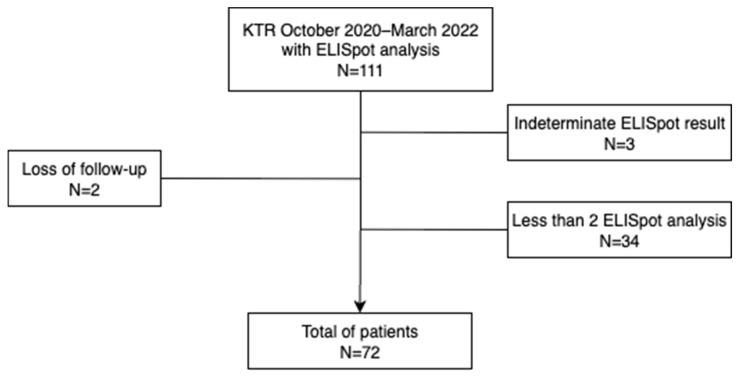
Diagram of patient enrollment and selection criteria.

**Figure 2 vaccines-13-00796-f002:**
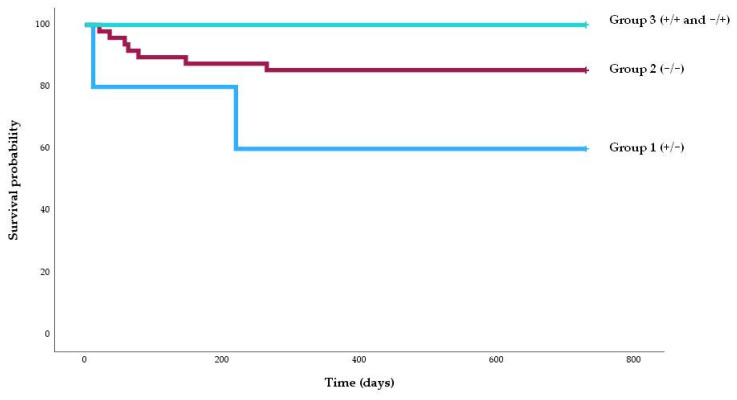
Kaplan–Meier survival graph of the percentage of patients free of BKPyV nephropathy according to the risk group. The percentage of patients free of BKPyV nephropathy was higher among patients in group 3 (100%) than those in group 2 (91.7%) and group 1 (60%) (log-rank, *p* = 0.011).

**Figure 3 vaccines-13-00796-f003:**
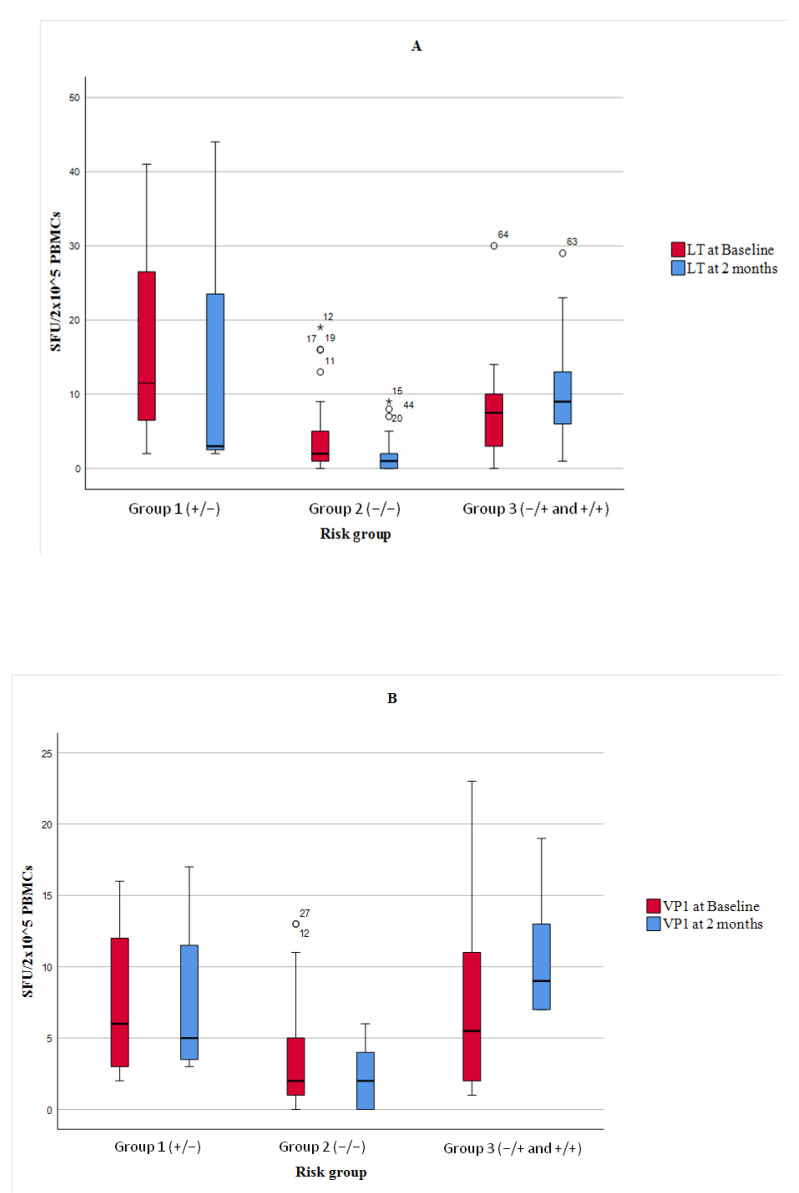
Box plots of BKPyV-specific T-cell responses measured by IFN-γ ELISpot using LT (Graph **A**) and VP1 (Graph **B**) antigens at baseline and two months post-transplant, stratified by ELISpot-defined risk groups. Group 1 = patients with a positive ELISpot pre-transplant that became negative; Group 2 = patients with negative ELISpot at both time points; Group 3 = patients with a positive ELISPOT post-transplant (−/+ or +/+). Results are expressed as spot-forming units (SFU) per 2 × 10^5^ PBMCs. Horizontal lines indicate medians, boxes represent IQRs, and whiskers show minimum and maximum values (excluding outliers *).

**Figure 4 vaccines-13-00796-f004:**
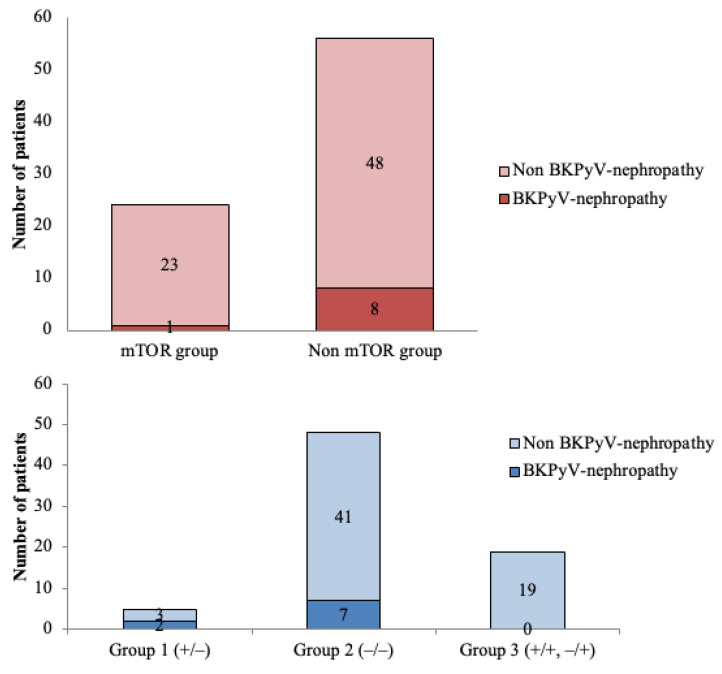
Incidence of BKPyV-nephropathy according to risk groups and mTOR-based immunosuppressive regimen. In the logistic regression model, patients with mTOR-based regimens presented less BKPyV-nephropathy compared with the non-mTOR-based regimens (*p* = 0.014). Furthermore, a statistically significant difference was observed in terms of BKPyV-nephropathy within the three different risk groups, showing the highest risk for group 1 and the lowest for group 3 (*p* = 0.016).

**Figure 5 vaccines-13-00796-f005:**
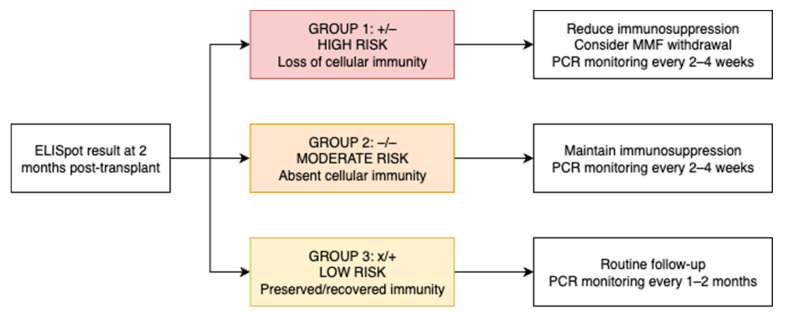
Proposed clinical algorithm for patient management based on ELISpot-defined BKPyV-specific T-cell response groups at 2 months post-transplant.

**Table 1 vaccines-13-00796-t001:** Main characteristics of the kidney transplant recipients.

	Total (n = 72)
**Age, median of years (SD)**	56.8 (13.6)
**Male sex, n (%)**	44 (61)
**Previous transplant, n (%)**	16 (22)
**First kidney allograft, n (%)**	55 (77)
**Renopancreatic transplantation, n (%)**	7 (10)
**Chronic kidney disease etiology**	
Diabetic nephropathy, n (%)	21 (29)
Polycystic kidney disease, n (%)	5 (7)
Nephroangiosclerosis, n (%)	4 (6)
Obstructive uropathy, n (%)	3 (4)
Other or undetermined, n (%)	39 (54)
**Deceased donation, n (%)**	62 (86)
**HIV infection, n (%)**	3 (4)
**Previous dialysis, n (%)**	58 (81)
**Immunosuppressive regimen before transplantation, n (%)**	9 (12.5)
**Immunosuppression regimen**	
Calcineurin inhibitor-based regimen, n (%)	47 (65)
mTOR-based regimen, n (%)	25 (35)
**Acute cellular rejection, n (%)**	15 (21)
**Creatinine at 1 month, mg/dL (SD)**	2 (1.6)
**Creatinine at 6 months, mg/dL (SD)**	1.6 (0.8)
**Creatinine at 1 year, mg/dL (SD)**	1.6 (0.7)
**Total lymphocytes at 2 months, /mm^3^ (SD)**	1340.8 (920)
**Total CD4-cells at 2 months (n = 54), /mm^3^ (SD)**	525.9 (486.3)
**BKPyV-DNAemia, n (%)**	9 (12.5)
**Median time to the first detection, days (SD)**	63 (148)
**Biopsy-proven BKPyV-nephropathy, n (%)**	3 (4)
**Cytomegalovirus D+/R-, n (%)**	13 (18)
**Cytomegalovirus infection, n (%)**	34 (47)
**Simultaneous BKPyV-CMV, n (%)**	4 (5)
**PRA, n (%)**	
0–10%	47 (65)
11–50%	9 (12.5)
>50%	16 (22)
**Presence of DSA, n (%)**	14 (19.5%)

Abbreviations: DSA: Donor-Specific Antibodies; PRA: Panel Reactive Antibodies; SD: Standard Deviation.

**Table 2 vaccines-13-00796-t002:** Logistic regression model of variables evaluated as predictive factors for BKPyV-nephropathy in kidney transplant recipients.

		Patients	BKPyV-Nephropathy	Univariate Analysis		Multivariate Analysis	
	Category	n	n (%)	OR (95% CI)	*p* Value	OR (95% CI)	*p* Value
Age	Age ≥ 60 Age < 60	31 41	3 (10) 6 (15)	0.6 (0.1–2.7)	0.53	1.8 (0.3–12.5)	0.5
Sex	Male Female	44 28	3 (7) 6 (21)	3.7 (0.8–16.3)	0.08	10.7 (1.2–96)	0.07
Previous transplant	Yes No	16 56	2 (12) 7 (12)	1 (0.2–5.3)	1		
First kidney allograft	Yes No	15 48	2 (13) 7 (15)	0.8 (1.2–3.3)	0.7		
Renopancreatic transplant	Yes No	7 65	1 (14) 8 (12)	1 (0.1–11)	0.9		
Immunosuppressive regimen before transplant	Yes No	9 63	1 (11) 8 (13)	0.8 (0.1–7.8)	0.9		
Use of lymphocyte-depleting agents	Yes No	36 36	4 (11) 5 (14)				
Maintenance immunosuppressive therapy	CI based mTOR based	48 24	8 (17) 1 (4)	0.2 (0.02–1.8)	0.16	0.02 (0.01–0.46)	0.014
Acute cellular rejection (3 months)	Yes No	15 57	3 (20) 6 (10)	2 (0.4–9.7)	0.3		
Lymphopenia (<1000/mm^3^)	Yes No	30 42	2 (7) 7 (17)	0.3 (0.06–1.8)	0.22	0.4 (0.9–1.2)	0.8
Cytomegalovirus D+/R-	Yes No	13 59	1 (7) 8 (14)	0.5 (0.05–4.5)	0.6		
Cytomegalovirus infection	Yes No	34 38	4 (12) 5 (13)	0.9 (0.2–3.5)	0.9		
ELISpot risk category	Group 1 (+/−)	5	2 (40)				
	Group 2 (−/−)	48	7 (15)				
	Group 3 (x/+)	19	0	6.4 (1.3–30.7)	0.02	19 (1.7–208)	0.016

## Data Availability

The data that support the findings of this study are available from the corresponding author [MB] upon reasonable request.

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
