# Peer review of "Utility of the ELISpot Test to Predict the Risk of Developing BK Polyomavirus Nephropathy in Kidney Recipients, a Multicenter Study"

_vaccines, 2025, doi:10.3390/vaccines13080796_

Round 1
Reviewer 1 Report
Comments and Suggestions for Authors
The paper by Sempere et al. describes the use of Elispot assays to predict the risk of BKVN. The paper indicates that such Elispot assays used in pre and post graft are of interest for the follow up of KT transplants. The paper is well written; nevertheless, I have some minor recommendations
P2 L45 consider “In healthy individuals BKPyV is asymptomatic”
P4 L168, correct the typo specific
P5 T1, what was/were the cause(s) of the unsuccess of initial transplants?
P6 L228, can you clarify “without BKPyV infection”.
P7 L236, the same for CMV
P7 L251, the sentence can be moved in the part dealing with mTOR results
P7 Two peptides pools have been use to assess T and VP1 IFN response, does the authors can bring recommendations based on sensitivity-specificity-ppv
P8, in line with previous comment, consider to move figure 2 after fig 4. Consider adding a legend to Y axis
Fig 3 consider adding the numbers at risk under the plots
P12 L351 The part on mTOR is rather long
Author Response
-
P2 L45: consider “In healthy individuals BKPyV is asymptomatic”
- The sentence has been corrected to: “In healthy individuals, BKPyV is asymptomatic.”
-
P4 L168: correct the typo “specific”
- Typo corrected ("specifc" → "specific").
-
P5 T1: what was/were the cause(s) of the unsuccess of initial transplants?
- We have now clarified the causes of previous graft loss in the 17 re-transplanted patients: 10 due to chronic cellular rejection (including 2 with thrombotic microangiopathy), 2 due to chronic humoral rejection, and 5 with mixed rejection.
-
P6 L228: can you clarify “without BKPyV infection”?
- We clarified “without BKPyV infection” as “without detectable BKPyV-DNAemia.”
-
P7 L236: the same for CMV
- Clarified CMV infection as “CMV-DNAemia detected”
-
P7 L251: the sentence can be moved in the part dealing with mTOR results
- The sentence regarding mTOR results has been moved to the paragraph where mTOR is discussed.
-
P7: Two peptides pools have been used to assess T and VP1 IFN response, do the authors bring recommendations based on sensitivity-specificity-PPV?
- The sensitivity and specificity of the IFN-γ ELISpot assay are high when using the most immunogenic peptides, such as LT and VP1, allowing the detection of 1 antigen-specific T cell per 100,000–300,000 PBMCs. While sensitivity could be further increased by evaluating a broader range of antigens, this would require a higher number of PBMCs and result in increased costs. Given the need for methods that are feasible in routine clinical practice and cost-effective, we propose IFN-γ ELISpot using LT and VP1 antigens of BK virus as the most suitable approach.
-
P8: in line with previous comment, consider to move Figure 2 after Fig 4. Consider adding a legend to Y axis
- We moved Figure 2 after Figure 4 as suggested and added a Y-axis legend.
-
P12 L351: The part on mTOR is rather long
- The mTOR discussion section has been shortened while keeping key references. We have restructured the paragraph to improve clarity and focus on the main findings.
Reviewer 2 Report
Comments and Suggestions for Authors
Suggestion for the Authors:
While the article is of high merit, a brief acknowledgment of this limitation in the "Limitations" section of the Discussion would further strengthen the manuscript. The authors could note that despite its high predictive utility, the cost and complexity of the ELISpot assay may limit its widespread adoption in resource-limited health systems, and that cost-effectiveness studies would be an important next step to justify its large-scale implementation.
Author Response
We appreciate your positive feedback and helpful suggestion. As requested, we have included the following in the Limitations section of the Discussion:
"However, it is important to emphasize the need for methods that support clinical decision-making while considering feasibility and cost-effectiveness. Including all viral antigens would likely be impractical, especially given the limited availability of PBMCs in these patients. Despite its promising predictive value, the ELISpot assay's cost and complexity may limit its implementation in resource-limited settings. Future cost-effectiveness studies are needed to support its broader clinical adoption.”
Reviewer 3 Report
Comments and Suggestions for Authors
The study analyzes monitoring BKPyV in a small KTx cohort (N = 72). Nine patients developed BKPyV nephropathy, and three had biopsy-proven disease.
Novelty is missing. The analyses are primarily negative, resulting from undepover analysis.
Any conclusion can be drawn from the study.
A nationwide perspective (at least) is needed with the analysis of decisions taken based on the diagnosis of BKPyV nephropathy, with the follow-up of their consequences.
Author Response
We thank the reviewer for the critical and valuable comments.
We acknowledge the limited sample size as a constraint and have added a comparative analysis between included and excluded patients to address potential selection bias. Despite the sample, the findings suggest trends that may guide future larger-scale studies. To our knowledge, this is one of the few prospective studies assessing pre- and post-transplant ELISpot changes with clinical endpoints. While confirmatory studies are needed, we believe the proposed ELISpot-based stratification algorithm offers a clinically useful approach.
We agree that a national, multicenter study including decision pathways and clinical outcomes would be highly valuable. However, this study was designed as a preliminary step to explore the feasibility and utility of ELISpot-guided stratification. Future multicenter research should build on these findings to assess decision-making strategies and long-term outcomes.
Round 2
Reviewer 3 Report
Comments and Suggestions for Authors
The explanation provided does not change the paper's content. The sample size is small. A two-center study cannot add new knowledge in the context of BKV nephropathy.
Please look at this more critically - 'Nine patients developed presumptive BKPyV-nephropathy, and three had biopsy-proven disease.'
The only message is 'A change in ELISpot response from positive to negative at 2 months post-transplant was associated with increased risk of BKPyV-nephropathy (p=0.016).'
Perhaps the authors should focus more on optimal testing time and explain how pre-transplant testing affects post-transplant results. These necessitate a different presentation of results and modification of the discussion.
Author Response
Dear Reviewer, Thank you for your critical assessment and for taking the time to review our manuscript. While we acknowledge your concerns, we respectfully believe that our study provides clinically meaningful contributions and novel insights into the risk stratification and management of BK polyomavirus (BKPyV) nephropathy.-
Prospective and updated evidence in a neglected area
BKPyV remains a highly prevalent and clinically relevant infection in kidney transplant recipients, with no approved antiviral treatment. Most ELISpot-based studies in this area are over a decade old. Our study provides contemporary, prospective, multicenter data, collected under current immunosuppressive protocols and supported by a standardized follow-up schedule. This reinforces the validity and applicability of our findings in real-life settings. -
Pragmatic and cost-effective immune-based monitoring strategy
Unlike previous studies that used multiple post-transplant ELISpot measurements, our approach requires only two time points (pre-transplant and 2 months post-transplant), providing a feasible, low-burden method to stratify patient risk. This is particularly relevant for centers with limited resources, as it avoids complex and frequent immune profiling. -
Adjustment of the algorithm to reflect personalized PCR monitoring
In response to your comment, we have updated the proposed algorithm (Figure 5) to reflect how ELISpot results could inform PCR surveillance intensity: - For moderate-risk patients (ELISpot -/-), we now recommend BKPyV-DNAemia monitoring every 2–4 weeks, rather than routine monthly testing.
-
For low-risk patients (ELISpot +/+ or -/+), who demonstrated a preserved virus-specific T-cell response, we suggest extending the interval to every 1–2 months during the first 9 months post-transplantation.
This adjustment offers a practical and cost-saving strategy that could reduce unnecessary testing in low-risk individuals while maintaining safety. -
Alignment with current trends in transplant virology (CMV 2025 guidelines)
Importantly, our strategy mirrors the evolving landscape of viral monitoring in transplant medicine. The 2025 International CMV Guidelines (Kotton et al.) now include T-cell immunity assays as part of clinical decision-making in high-risk CMV patients. We believe BKPyV may follow a similar path, and our study provides preliminary evidence supporting that shift. -
Sample size and power
We acknowledge the relatively modest sample size, as already stated in the limitations section. However, the study achieved a post-hoc power of 88.8% to detect clinically meaningful differences in ELISpot-defined risk groups. Moreover, the statistically significant association between loss of immunity and BKPyV-nephropathy (p=0.016), along with the protective effect of mTOR-based regimens, further supports the robustness of our findings. -
Added value beyond a single message
While you mention that the only message is the association between ELISpot changes and nephropathy, we respectfully highlight that our work also contributes: - A simplified, implementable clinical algorithm based on cellular immunity;
- A proposal to optimize PCR monitoring schedules according to immune status;
- A signal of the protective effect of mTOR-based immunosuppression, aligning with prior mechanistic studies;
- And a parallel with the direction of current CMV management, advocating for immunologically guided care.
Nevertheless, based on your suggestions, we revised all the manuscript to specify the aspects detailed above, placing greater emphasis on the optimal testing time and the proposed algorithm (please look at the marked parts).
Round 3
Reviewer 3 Report
Comments and Suggestions for Authors
The revised version is better. The lack of multicentre design is a drawback that cannot be improved.